# Chronic Plasma Exposure to Kinase Inhibitors in Patients with Oncogene-Addicted Non-Small Cell Lung Cancer

**DOI:** 10.3390/cancers12123758

**Published:** 2020-12-14

**Authors:** Arthur Geraud, Laura Mezquita, Edouard Auclin, David Combarel, Julia Delahousse, Paul Gougis, Christophe Massard, Cécile Jovelet, Caroline Caramella, Julien Adam, Charles Naltet, Pernelle Lavaud, Anas Gazzah, Ludovic Lacroix, Etienne Rouleau, Damien Vasseur, Olivier Mir, David Planchard, Angelo Paci, Benjamin Besse

**Affiliations:** 1Cancer Medicine Department, Gustave Roussy, 94805 Villejuif, France; arthur.geraud@gustaveroussy.fr (A.G.); lmezquitap@clinic.cat (L.M.); charles.naltet@gustaveroussy.fr (C.N.); pernelle.lavaud@gustaveroussy.fr (P.L.); david.planchard@gustaveroussy.fr (D.P.); 2Early Drug Development Department (DITEP), Gustave Roussy, 94805 Villejuif, France; christophe.massard@gustaveroussy.fr (C.M.); anas.gazzah@gustaveroussy.fr (A.G.); 3Translational Genomics and Targeted Therapeutics in Solid Tumors, August Pi i Sunyer Biomedical Research Institute (IDIBAPS), Medical Oncology Department, Hospital Clínic, 08036 Barcelona, Spain; 4Department of Medical and Digestive Oncology, Hôpital Européen Georges Pompidou, Assistance Publique des Hôpitaux de Paris, 75015 Paris, France; edouard.auclin@aphp.fr; 5Pharmacology Department, Gustave Roussy, 94805 Villejuif, France; david.combarel@gustaveroussy.fr (D.C.); julia.delahousse@gustaveroussy.fr (J.D.); angelo.paci@gustaveroussy.fr (A.P.); 6Faculty of Pharmacy, Paris-Saclay University, 92296 Chatenay-Malabry, France; 7Department of Pharmacology and Clinical Investigation Center, Pitié-Salpêtrière Hospital, INSERM, CIC-1421, Sorbonne University, 75013 Paris, France; paul.gougis@aphp.fr; 8CLIP2 Galilée, Regional Pharmacovigilance Center, Pitié-Salpêtrière Hospital, INSERM, CIC-1421, Sorbonne University, 75013 Paris, France; 9Paris-Saclay University, Cancer Campus Gustave Roussy, Gustave Roussy, 94805 Villejuif, France; 10Department of Medical Biology and Pathology, Gustave Roussy, 94805 Villejuif, France; cecile.jovelet@gustaveroussy.fr (C.J.); ludovic.lacroix@gustaveroussy.fr (L.L.); etienne.rouleau@gustaveroussy.fr (E.R.); damien.vasseur@gustaveroussy.fr (D.V.); 11Department of Medical Imaging, Gustave Roussy, 94805 Villejuif, France; caroline.caramella@gustaveroussy.fr; 12Pathology Department, Gustave Roussy, 94805 Villejuif, France; julien.adam@gustaveroussy.fr; 13Department of Ambulatory Cancer Care, Gustave Roussy, 94805 Villejuif, France; olivier.mir@gustaveroussy.fr

**Keywords:** non-small cell lung cancer, oncogene addiction, kinase inhibitors, chronic plasmatic exposure, resistance mutation

## Abstract

**Simple Summary:**

In this study, we measured the plasmatic concentration of Kinase inhibitors (KI) among a population with non-small cell lung cancer (NSCLC) harboring driver genetic alterations. They received erlotinib, gefitinib, osimertinib, crizotinib, or dabrafenib (with or without trametinib) for at least three months. The results were measured by ultra-performance liquid chromatography coupled with tandem mass spectrometry and compared to previously published data. Between November 2013 and February 2019, fifty-one samples were analyzed. The main outcome was the rate of samples with suboptimal KI plasma concentrations. Suboptimal plasma concentrations were observed in 51% (26/51) of cases and might contribute to treatment failure.

**Abstract:**

Kinase inhibitors (KI) have dramatically improved the outcome of treatment in patients with non-small cell lung cancer (NSCLC), which harbors an oncogene addiction. This study assesses KI plasma levels and their clinical relevance in patients chronically exposed to KIs. Plasma samples were collected in NSCLC patients receiving erlotinib, gefitinib, osimertinib, crizotinib, or dabrafenib (with or without trametinib) for at least three months between November 2013 and February 2019 in a single institution. KI drug concentrations were measured by ultra-performance liquid chromatography coupled with tandem mass spectrometry and compared to published data defining optimal plasma concentration. The main outcome was the rate of samples with suboptimal KI plasma concentrations. Secondary outcomes included its impact on *T790M* mutation emergence in patients receiving a first-generation epidermal growth factor receptor (EGFR) KI. Fifty-one samples were available from 41 patients with advanced NSCLC harboring driver genetic alterations, including *EGFR*, v-Raf murine sarcoma viral oncogene homolog B (*BRAF*), anaplastic lymphoma kinase (*ALK*) or ROS proto-oncogene 1 (*ROS1*), and who had an available evaluation of chronic KI plasma exposure. Suboptimal plasma concentrations were observed in 51% (26/51) of cases. In *EGFR*-mutant cases failing first-generation KIs, *EGFR* exon 20 p.T790M mutation emergence was detected in 31% (4/13) of samples in optimal vs. none in suboptimal concentration (0/5). Suboptimal plasma concentrations of KIs are frequent in advanced NSCLC patients treated with a KI for at least three months and might contribute to treatment failure.

## 1. Introduction

Kinase inhibitors (KIs) form the cornerstone of the therapeutic strategy in patients with non-small cell lung cancer (NSCLC) harboring molecular driver alterations, such as epidermal growth factor receptor (*EGFR*) mutations, v-Raf murine sarcoma viral oncogene homolog B (*BRAF*) mutations, anaplastic lymphoma kinase (*ALK*) fusions, ROS proto-oncogene 1 (*ROS1*) fusions, etc. [1]. Although an initial benefit under KIs is common, all patients ultimately experience disease progression. One cause of KI failure is the acquisition of resistance mechanisms at the molecular level. Next-generation KIs were successfully developed to overcome resistance to the previous-generation KIs. One of the best-known examples is osimertinib, which has demonstrated impressive antitumor activity against NSCLC harboring an *EGFR* mutation, either an exon 19 deletion (Ex19del), an exon 21 p.L858R, or an exon 20 p.T790M, the latter being acquired under first-generation kinase inhibitors (erlotinib, gefitinib) [2]. However, few KIs have been approved in NSCLC, and strategies aimed at optimizing the duration of response are needed. One such approach involves a better understanding of the pharmacology of KIs.

In oncology, suboptimal KI plasma concentrations have been associated with a lack of antitumor efficacy (Appendix A) [3,4,5], and represents a plausible explanation for KI failure. The therapeutic index of this category of drugs is often narrow, and most are prescribed as a flat dose. Furthermore, KI plasma concentrations vary depending on several factors, such as body weight, smoking status [6], concomitant drug intake [7,8], and adherence [9]. Plasma concentrations of KIs can also decrease over time, due to the drug’s pharmaceutical properties, such as CYP3A5/CYP3A4 auto-induction or P-glycoprotein auto-induction, which will increase the drug’s clearance [10].

For KIs, the pharmacokinetic steady state is usually reached after two to three weeks of treatment depending on the half-life of the drug, however, data on the dynamic evolution of KI plasma concentrations are limited [5]. A retrospective study of plasma concentration of sorafenib measured every 15 days by liquid chromatography in a population of hepatocellular carcinoma patients has been reported, with sampling from initiation of treatment until progression [10]. This showed that exposure after three months of treatment was lower than after one month of treatment. In the event of a decrease in plasma concentration over time, intra-patient adaptation could be discussed within the framework of target drug monitoring (TDM) [4]. TDM involves the measurement and interpretation of drug concentrations in biological fluids to determine drug dosage for an individual patient. This strategy is used in clinical practice for various therapeutic drug classes, including antibiotics, immunosuppressors, antiepileptics, and antiarrhythmic agents [4,11]. In oncology, TDM is widely accepted for imatinib in chronic myeloid leukemia [12], and is currently being explored in oncogene-addicted NSCLC [3,4,5]. Good candidates for TDM are drugs with a narrow therapeutic window and a direct relationship between plasma concentrations and efficacy or toxicity pharmacokinetic-pharmacodynamic relationships [13]. In oncogene-addicted advanced NSCLC, chronic plasma exposure to KIs and its clinical relevance, in particular at the time of disease progression, is poorly described. We aimed to assess the clinical and molecular impact of chronic suboptimal KI plasma concentrations in patients with oncogene-addicted advanced NSCLC.

## 2. Patients and Methods

A monocentric observational study was performed in the context of routine clinical care.

### 2.1. Patients

Patients with advanced NSCLC harboring oncogenic drivers, such as an *EGFR-*mutation, *BRAF^V600E^*-mutation, *ALK*-fusion, or *ROS1-*fusion, treated with KI therapy for at least three months were eligible. Patients receiving erlotinib, gefitinib, osimertinib, crizotinib, or dabrafenib (with or without trametinib) in the context of a trial (academic or clinical), expanded access, or routine clinical care between November 2013 and February 2019 were screened. KI plasma concentrations were evaluated in blood samples collected during therapy, and at the time of clinical response or relapse. Chronic plasma exposure was defined as at least three months of KI therapy. Clinical and pathological data were extracted from electronic medical records. Radiological assessments were performed every 8 or 12 weeks per Response Evaluation Criteria in Solid Tumors (RECIST) v1.1 and per the treating physician’s discretion.

### 2.2. Assessment of Plasma Exposure

Residual plasma concentrations of drugs were measured in blood samples using ultra-performance liquid chromatography coupled with tandem mass spectrometry validated methods [10]. A residual estimate based on pharmacokinetic and population pharmacokinetic analysis parameters (Appendix A) was made for samples not collected at the time of the residual sampling.

Additional information about the methodology is present in the Appendix A.

Precise measurement of trough concentrations is challenging in routine practice, due to the constraints of the logistics of blood collection at a precise time and the fact that the timing of administration varies from patient to patient. We defined three situations according to the dosing time with respect to the last dose: Optimal, evaluable, and non-interpretable.

**Optimal:** The optimal concentration corresponds to the true residual plasma concentration at a steady state, in the blood collection performed immediately before the next administration.**Evaluable:** The residual plasma concentration is estimated by an extrapolation method from known pharmacokinetic parameters (distribution volume, half-life, clearance) and from data obtained in population pharmacokinetic models [14]. This estimate of standard trough concentration (C_min_, std) is only possible when blood samples were collected at steady state or during the terminal elimination phase of the drug, since in this phase, the elimination rate is linear [14,15].
-C (min, std) = C(t) * 0.5 ^ (Delta (t)/t1/2)-C (min, std) = C(t) * exp (k(e) x Delta (t))

Delta t = t − tau, tau is 24 h when collecting a sample once a day, or 12 h when collecting samples twice a day, k(e) is the elimination rate constant.

Pharmacokinetic parameters and a population pharmacokinetic study are summarized in Appendix A.

3.**Not interpretable:** Extrapolation is not feasible for samples taken during the plasma peak period.

Residual plasma concentrations (optimal and evaluable) were compared to target values recommended in the literature for TDM to define the ‘optimal concentration’ for each KI (erlotinib > 500 ng/mL, gefitinib > 200 ng/mL, osimertinib > 166 ng/mL, crizotinib > 235 ng/mL, dabrafenib > 96.1 ng/mL, trametinib > 10.6 ng/mL) or ‘suboptimal concentration’ [3,4,5] (Appendix A).

### 2.3. Somatic Molecular Analysis

*ALK* fusion was assessed by immunohistochemistry (IHC) or fluorescence in situ hybridization (FISH), while *ROS1* FISH was used to assessed fusions, and reverse transcriptase-polymerase chain reaction (RT-PCR) or next-generation sequencing (NGS) was used to assess mutations and/or other alterations.

Blood sample collections and ctDNA (for mutational analyses) were collected over time, including at the time of radiological disease evaluation. Plasma was isolated, and ctDNA analysis was performed centrally (Gustave Roussy, France) using a targeted panel. The panel used is described in Table 1.

Panels tested in blood samples (liquid biopsy):-CHP2 (Ion AmpliSeq Cancer Hotspot Panel v2 (CHP2)) designed to amplify 207 amplicons covering 50 genes (Thermo Fisher Scientific)).-Oncomine lung (Oncomine Lung ctDNA Assay contains 35 amplicons covering 11 genes (Thermo Fisher Scientific)). A unique molecular identifier is combined with each single DNA molecule. For calling the detected variants, the following parameters were applied: Allele Read count > 10; Fusion read count > 1; Variant type: Single nucleotide variant, insertion-deletion, multi-nucleotide variant, copy number variant, long deletion, fusion; Variant effect: Unknown, missense, none frameshift Insertion, none frameshift Deletion, non-sense, stop loss, frameshift insertion, frameshift deletion.


Panels used for tissue samples:
-MOSC4 (Ion AmpliSeq MOSC4 designed to cover 82 genes, combining two other panels (CHP2 + Safir02)).-OCAV3 (Ion AmpliSeq Oncomine Comprehensive Assay V3 enables the detection of mutations across 161 genes, gene fusions, and copy number variations (Thermo Fisher Scientific)).-Sentosa SQ NSCLC (Sentosa SQ Non-Small Cell Lung Cancer panel targets 11 genes with 28 amplicons (Vela Diagnostics)). For calling the detected variants, following parameters were applied: 5000 Exomes Global MAF < 0.01; Allele frequency > 0.02; Allele ratio > 0.02; Allele read counts > 50; Alternate read counts > 30; Fusion read counts > 50; Variant effect: Unknown, missense, none frameshift insertion, none frameshift deletion, stop loss, non-sense, frameshift insertion, frameshift deletion; Variant type: Single nucleotide variant, insertion-deletion, multi-nucleotide variant, copy number variant, long deletion, fusion.

### 2.4. Statistical Analysis

Median values (interquartile range) and frequencies (percentage) were calculated for continuous and categorical variables, respectively. Medians and proportions were compared using the Student’s *t*-test and chi-square test (or Fisher’s exact test, if appropriate), respectively.

Time to treatment failure was defined as the time between KI initiation and progression. Overall survival (OS) was defined as the time between KI initiation and death from any cause. Time to treatment failure and OS were estimated using the Kaplan-Meier method and described using median values with their 95% confidence intervals (95% CI). Follow-up was calculated using the reverse Kaplan-Meier method. Correlation between exposure and *EGFR* exon 20 p.T790M mutation occurrence was evaluated with a Pearson correlation coefficient.

All statistical analyses were performed with R studio version 2.15.2, *p*-values < 0.05 were considered statistically significant, and all tests were two-sided.

## 3. Results

A total of 94 plasma samples from 71 patients were prospectively collected. Among them, 43 samples were excluded, due to missing data, including the time of last drug intake (*n* = 21), technical issues (*n* = 13), and treatment for less than three months (*n* = 9) (Figure 1). A total of 51 samples from 41 patients were eligible for evaluation. During the evaluation period, patients received treatment with erlotinib (*n* = 13), gefitinib (*n* = 11), osimertinib (*n* = 10), crizotinib (*n* = 7) and dabrafenib (*n* = 5) + trametinib (*n* = 5). Baseline characteristics are summarized in Table 2 for eligible samples and in Appendix A for patients. The 51 eligible samples were collected after a median of 20.3 months (95% CI, 6.29 to 25.5) on KI treatment.

### 3.1. Suboptimal Concentration and KI Response

Suboptimal KI plasma concentrations were observed in 26 (51%) samples from 20 (49%) patients. Clinical characteristics, according to KI concentration (suboptimal vs. optimal) are summarized in Table 2. No significant differences were observed for any characteristics evaluated, including tobacco consumption and other drug intake. For samples collected at the time of disease progression, the suboptimal plasma concentration rate was 43% (16/37) vs. 71% (10/14) in samples from patients without progression at the time of sample collection. In cases of isolated intracranial progression, suboptimal plasma concentrations were reported in 43% (3/7) of cases (Figure 2).

### 3.2. Clinical Relevance of Suboptimal Concentration

After a median follow-up of 52.4 months (95% CI 35.6 to 106.9), the median OS was 61.1 months (95% CI 30.6 to not reached). The median time to treatment failure in the overall population was 15.9 months (95% CI 13.7 to 25.8). In patients with suboptimal concentrations (*n* = 26), time to treatment failure was 14.2 months (95% CI 12.48 to 57.4) vs. 18.1 months (95% CI 8.65 to 58.9) in the optimal concentration group (*n* = 25; *p* = 0.9) (Appendix A). No significant difference was observed according to the type of KI administered in optimal vs. suboptimal concentration groups (log-rank test, *p* = 0.52) (Table 2). At the time of progression (*n* = 16), the median time to treatment failure in the suboptimal concentration group was 16.2 months (95% CI 7.8 to 27.2) vs. 11.4 months (95% CI 7.4 to 57.4) in the optimal concentration, which was not significantly different (log-rank test, *p* = 0.8) (Figure 3).

### 3.3. KI Exposure and Resistance Mechanisms

Among patients with *EGFR*-mutated NSCLC failing a first-generation KI (*n* = 24), 18 patients had a molecular evaluation to assess resistance mechanisms. The emergence of *EGFR* exon 20 p.T790M mutations was detected in 31% (4/13) of patients in the optimal concentration group vs. none in the suboptimal concentration group (0/5) (Spearman r = −0.33, *p* = 0.18) (Table 1).

## 4. Discussion

In our cohort of patients with oncogene-addicted NSCLC treated with a KI for at least three months, 51% of blood samples showed suboptimal KI concentrations (plasma concentration below the published recommended values). This is consistent with studies of KI concentrations for other cancers, which report suboptimal concentration rates ranging from 11% to 83% with KIs, including 11% for erlotinib, 49% for sunitinib, and 65–73% for imatinib [12,16]. Various hypotheses have been formulated to explain decreasing concentrations over time, such as a quantitative decrease, due to intrinsic pharmacological properties (CYP3A4 auto-induction, P-glycoprotein auto-induction, etc.), poor adherence, and drug–drug interactions [7,9,10]. In our cohort, 32% of patients (*n* = 13) received a proton pump inhibitor and 12% (*n* = 4) were current smokers. These two parameters had no measurable effect on plasmatic exposure. However, cigarette smoke induces CYP1A1 and significantly decreases erlotinib plasmatic exposure, but has no known effect on the exposure of KIs metabolized mostly by other cytochromes (CYP3A4, CYP3A4, CYP2C8, among others) [6]. The absorption of erlotinib and gefitinib requires gastric acidity. Proton pump inhibitors are known to interact with erlotinib and gefitinib, decreasing their bioavailability and the plasmatic exposure [7,17]. This interaction is not relevant for crizotinib, dabrafenib, osimertinib, and trametinib [7,17]. Furthermore, in our study, data were collected retrospectively, making an objective evaluation of adherence impossible, which is one of the main causes of low-plasmatic exposure [9,18].

Suboptimal concentrations could contribute to treatment failure with KIs. The median time to treatment failure was lower in the suboptimal group vs. the optimal concentration group (14.2 vs. 18.2 months), although this was not statistically significant, possibly due to the small number of cases. We also observed a 43% rate of suboptimal concentrations in patients with isolated intracranial progression, which is relevant in light of the low ability of some KIs to cross the blood-brain-barrier, resulting in even lower intracranial concentrations [19].

In patients failing a first-generation EGFR KI, the emergence of the resistance *EGFR* exon 20 p.T790M is expected in half of these cases [20]. In our cohort, an *EGFR* exon 20 p.T790M mutation was not detected in any of the five patients with suboptimal concentrations vs. 31% of patients in the optimal concentration group. Although the sample size was small, this supports the hypothesis that tumor progression is more likely with an insufficient plasma concentration of the drug before acquiring molecular resistance. Exclusion of concomitant interfering drugs or intra-patient dose escalation could be efficient strategies for overcoming this phenomenon.

This approach of including pharmacological evaluations and intra-patient dose adaptation in a TDM strategy, to restore plasmatic exposure and possibly tumor response has not been evaluated in oncogene-addicted NSCLC. However, it has been proposed in other malignancies, such as chronic myeloid leukemia with imatinib [4,12,13], thyroid cancer with sorafenib [21], and renal cancer with axitinib [22].

This study has various limitations, including missing data and incomplete adherence data, due to the retrospective nature of the clinical data collection. The sample size is limited, and the population was heterogeneous and received different KIs. In addition, there was no evaluation of the concentration evolution over time that could explain the high rate of suboptimal concentrations in the clinical benefit group (Figure 2). Thus, these findings should be validated in larger prospective studies where different molecular populations are well represented. This may confirm that low plasmatic exposure at tumor progression correlates with KI failure and the emergence of resistance mutations.

## 5. Conclusions

KI suboptimal concentrations were observed in approximately 50% of advanced NSCLC patients with chronic exposure in our institution; the emergence of resistance mutations was only seen in optimal concentration cases, supporting the hypothesis of suboptimal concentrations as a potential explanation for KI failure.

## Figures and Tables

**Figure 1 cancers-12-03758-f001:**
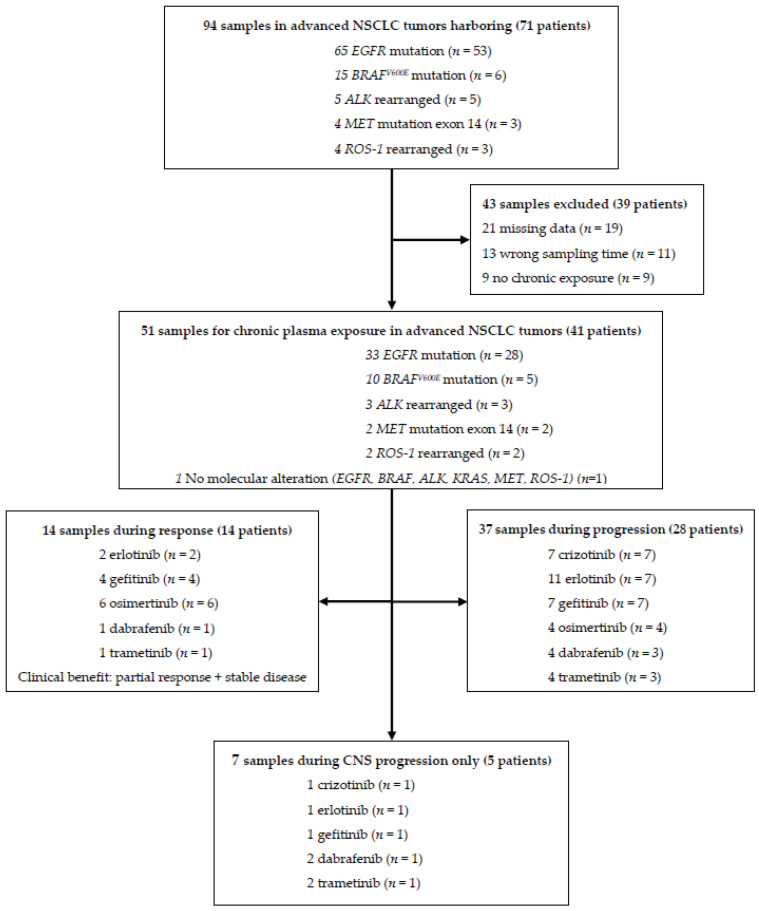
Flow diagram of samples and the study population.

**Figure 2 cancers-12-03758-f002:**
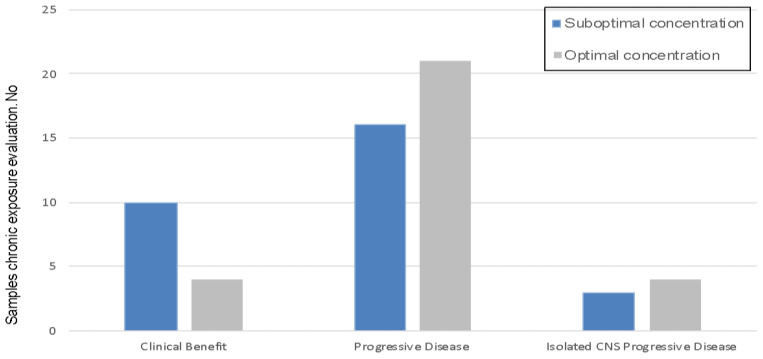
Distribution of plasma exposures according to clinical outcome. Clinical Benefit: Stable disease and partial response. Isolated CNS PD: Isolated central nervous system progressive disease.

**Figure 3 cancers-12-03758-f003:**
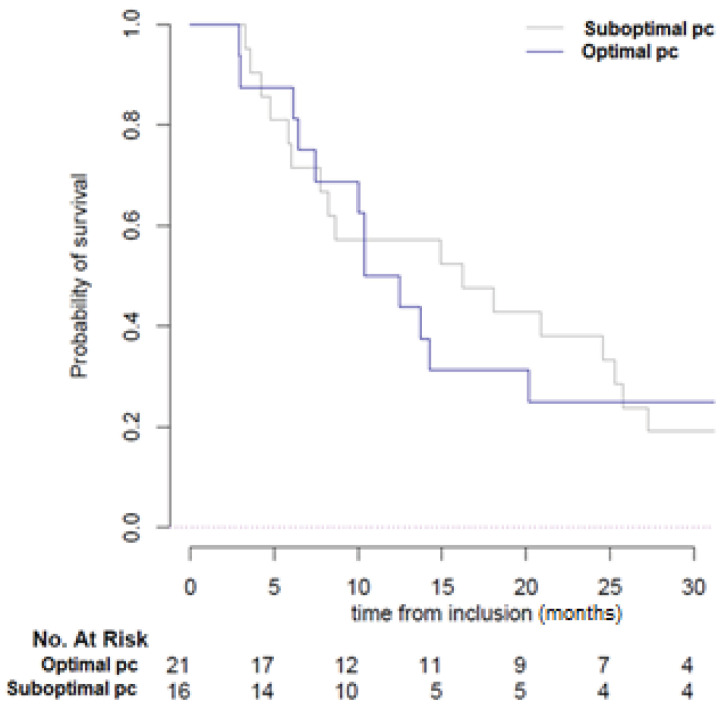
Kaplan-Meier curves of time to treatment failure according to KI plasma concentration. Sub-optimal pc: Suboptimal plasma concentration, Optimal pc: Sub-optimal plasma concentration.

**Table 1 cancers-12-03758-t001:** The molecular testing panel in tissue/blood samples from 18 patients with *EGFR*-mutated NSCLC after failing a first-generation kinase inhibitor.

N#	Driver Alteration	Other Alteration	Kinase Inhibitor	NGS Panel (Tissue)	NGS Panel (Blood)	*EGFR T790M* ddPCR Stilla^®^ (Blood)
1	*EGFR* exon 21 p. L858R	No	Gefitinib	-	CHP2 *: Not detected	Not detected
2	*EGFR* exon 21 p. L858R	No	Gefitinib	-	-	*T790M* detected
3	*EGFR* exon 21 p. L858R	No	Gefitinib	Oncomine^®^ Lung: *T790M* detected	CHP2 *: Not detected	-
4	*EGFR* exon 21 p. L858R	No	Gefitinib	-	CHP2 *: Not detected	Not detected
5	*EGFR* exon 19 deletion	TP53	Gefitinib	MOSC4: *T790M* not detected	-	-
6	*EGFR* exon 19 deletion	No	Gefitinib	-	-	Not detected
7	*EGFR* exon 19 deletion	No	Gefitinib	MOSC4: *T790M* not detected	-	*T790M* detected
8	*EGFR* exon 21 p.L833F	*TP53, CKDN2A*	Erlotinib	-	-	ddPCR Not detected
9	*EGFR* exon 19 deletion	No	Erlotinib	-	-	Not detected
10	*EGFR* exon 19 deletion	No	Erlotinib	-	-	Not detected
11	*EGFR* exon 19 deletion	No	Erlotinib	-	-	Not detected
12	*EGFR* exon 19 deletion	No	Erlotinib	MOSC4: *T790M* detected	-	*T790M* detected
13	*EGFR* exon 21 p. L858R	No	Erlotinib	Unknown: not detected	-	-
14	*EGFR* exon 19 deletion	No	Erlotinib	MOSC4: *T790M* not detected	-	Not detected
15	*EGFR* exon 19 deletion	No	Erlotinib	MOSC4: *T790M* not detected	-	-
16	*EGFR* exon 19 deletion	No	Erlotinib	MOSC4: *T790M* not detected	CHP2 *: Not detected	-
17	*EGFR* exon 21 p. L858R	No	Erlotinib	OCAV3: *T790M* not detected	Oncomine^®^ Lung: Not detected	-
18	*EGFR* exon 19 deletion	No	Erlotinib	NSCLC: *T790M* not detected	-	Not detected

(-): Test not performed; ddPCR: droplet digital polymerase chain reaction, CHP2 *: Ion Ampliseq Cancer Hotspot Panel v2 targeting 50 cancer genes (Thermo Fisher Scientific), MOSC4: Gustave Roussy homemade panel targeting 82 genes, OCAV3: Oncomine Comprehensive Assay v3M (Thermo Fisher Scientific) with 161 genes and NSCLC: Sentosa SQ Non-Small Cell Lung Cancer Panel (Vela Diagnostics) targeting 11 genes.

**Table 2 cancers-12-03758-t002:** Sample baseline characteristics and according to plasma kinase inhibitor concentration.

Characteristics, No. (%)	All Samples (*n* = 51)	Optimal Concentration (*n* = 25)	Suboptimal Concentration (*n* = 26)	*p*-Value
**Molecular alteration:***ALK*-rearranged, *BRAF^V600E^* mutation, *EGFR* deletion exon 19, *EGFR* mutation exon 21 (L858R), *EGFR* mutation exon 21 (L833F), *ROS-1* rearranged, *MET* mutation exon 14, No molecular alteration *(ALK, BRAF, EGFR, KRAS, MET, ROS-1*)	3 (7%), 10 (19%), 23 (45%), 9 (17%), 1 (2%), 2 (4%), 2 (4%), 1 (2%)	0 (0%), 4 (16%), 12 (48%), 6 (24%), 1 (4%), 1 (4%), 1 (4%), 0	3 (12%), 6 (22%), 11 (42%), 3 (12%), 0, 1 (4%), 1 (4%), 1 (4%)	*p* = 0.50
**Kinase inhibitor:** Erlotinib, Gefitinib, Osimertinib, Crizotinib, Dabrafenib, Trametinib	13 (26%), 11 (21%), 10 (19%), 7 (14%), 5 (10%), 5 (10%)	7 (28%), 8 (32%), 4 (16%), 2 (8%), 2 (8%), 2 (8%)	6 (23%), 3 (11.5%), 6 (23%), 5 (19.5%), 3 (11.5%), 3 (11.5%)	*p* = 0.52
**Stage at diagnosis:** I-II, III, IV	1 (2%), 2 (4%), 48 (94%)	0, 2 (8%), 23 (92%)	1 (4%), 0, 25 (96%)	*p* = 0.24
**Lines of previous kinase inhibitors:** ≤2, >2	44 (86%), 7 (14%)	20 (80%), 5 (20%)	24 (92%), 2 (8%)	*p* = 0.25
**Current smoker:** Yes, No	5 (10%), 46 (90%)	1 (4%), 24 (96%)	4 (15%), 22 (85%)	*p* = 0.35
**Concomitant proton pump inhibitor:** No, Yes	35 (69%), 16 (31%)	15 (60%), 10 (40%)	20 (77%), 6 (23%)	*p* = 0.19

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
