# Peer review of "Chronic Plasma Exposure to Kinase Inhibitors in Patients with Oncogene-Addicted Non-Small Cell Lung Cancer"

_cancers, 2020, doi:10.3390/cancers12123758_

Round 1

Reviewer 1 Report

The authors have addressed my concerns in this revision. In my opinion, the revised manuscript is acceptable for publication.

Reviewer 2 Report

I have no further comments.

Reviewer 3 Report

nA

This manuscript is a resubmission of an earlier submission. The following is a list of the peer review reports and author responses from that submission.

Round 1

Reviewer 1 Report

Comments:
1. The manuscript should be polished by a native English speaker as there are many awkward phrasings and grammatical errors. For example,
Line 35: “Secondary outcomes was” should be “Secondary outcome was”
Line 38: “Suboptimal concentration were observed” should be “Suboptimal concentration was observed”
Line 40: “Suboptimal plasma concentration are” should be “Suboptimal plasma concentration is”

2. All the abbreviations, such as NSCLC, EGFR, BRAF, ALK and ROS1 should be defined at its first mention in the manuscript. Perhaps the authors know the full names of these acronyms, but it should be assumed that the readers of the article may be students or young scientists.
3. I recommend the introduction should be intensively improved since it routinely introduces some background without adequate scientific depth. It is hard to understand the significance of this work from the presented introduction due to some key information is missing. For example, what is the main problem for recent kinase inhibitors? Why the study of the dynamic evolution of kinase inhibitors plasma concentrations is necessary? Why the data on the dynamic evolution of kinase inhibitors plasma concentrations are limited?
4. Table 1: The authors claimed n=26 for suboptimal concentration in the row of molecular alteration. However, the actual number is 25. Please double check it.
5. Figure 3: What is the unit of the x-axis? Week or month?
6. To highlight the significance of this work, the authors should adequately compare and discuss previously clinical data in this field rather than simply show their own results.

Reviewer 2 Report

The manuscript entitled "Chronic plasma exposure to kinase inhibitors in patients with oncogene-addicted non-small cell lung cancer" highlighted that suboptimal plasma concentration are frequent in advanced NSCLC patients treated with kinase inhibitors for at least three months and might contribute to treatment failure.

  • The Authors should perform a minor spell check in the text.
  • The Authors should provide the extensive forms for all acronyms, including gene acronyms, through the text when they first appear.
  • In the Methods section the Authors should report the parameters adopted to call the detected variants.
  • Mutations should be reported as follow: gene exon p.mutation (e.g. EGFR exon 20 p.T790M).

Reviewer 3 Report

Below are the few things authors need to address.

  1. The n is small for few molecular Alterations (example n-2 for ROS-1, MET, n-3 for ALK) to draw robust conclusions.
  2. What stages are these patients treated? And to what stage did the cancer advance to or decease after treatment? The response significantly varies to the patients that are treated at an advance stage compared patients treated at an early stage. Also if they are pretreated with other drugs and age. So detailed information of stage, pretreatment, age should be listed. All these effect conclusions/interpretations of plasm concentration to response. 
  3. It would be valuable addition to determine the reason for low plasma level of drugs, for example inactive CYP enzymes etc. factors